# The Effectiveness of Using a Pretrained Deep Learning Neural Networks for Object Classification in Underwater Video

**Piotr Szymak** [1,*,†] **, Paweł Piskur** [1,†] **and Krzysztof Naus** [2,†]

1   Faculty of Mechanical and Electrical Engineering, Polish Naval Academy, 80-127 Gdynia, Poland;
    p.piskur@amw.gdynia.pl
2   Faculty of Navigation and Naval Weapons, Polish Naval Academy, 80-127 Gdynia, Poland;
    k.naus@amw.gdynia.pl
*   Correspondence: p.szymak@amw.gdynia.pl
†   These authors contributed equally to this work.

**Abstract:** Video image processing and object classification using a Deep Learning Neural Network (DLNN) can significantly increase the autonomy of underwater vehicles. This paper describes the results of a project focused on using DLNN for Object Classification in Underwater Video (OCUV) implemented in a Biomimetic Underwater Vehicle (BUV). The BUV is intended to be used to detect underwater mines, explore shipwrecks or observe the process of corrosion of munitions abandoned on the seabed after World War II. Here, the pretrained DLNNs were used for classification of the following type of objects: fishes, underwater vehicles, divers and obstacles. The results of our research enabled us to estimate the effectiveness of using pretrained DLNNs for classification of different objects under the complex Baltic Sea environment. The Genetic Algorithm (GA) was used to establish tuning parameters of the DLNNs. Three different training methods were compared for AlexNet, then one training method was chosen for fifteen networks and the tests were provided with the description of the final results. The DLNNs were trained on servers with six medium class Graphics Processing Units (GPUs). Finally, the trained DLNN was implemented in the Nvidia JetsonTX2 platform installed on board of the BUV, and one of the network was verified in a real environment.

**Keywords:** object clasiffication in underwater video; deep learning neural network; genetic algorithm; AlexNet; DenseNet 201; GoogleNet; Inception ResNet v2; Inceptionv3; MobileNetV2; NASNet Mobile; ResNet 18; ResNet 50; ResNet 101; ShuffleNet; SqueezeNet; VGG 16; VGG 19; Xception

## 1. Introduction

Image recognition is becoming increasingly present in our daily lives, for example, in driver assistance systems [1], pedestrian location [2] or medical imaging systems [3,4] that can give a preliminary diagnosis from the image just like a human specialist would.

Image recognition can be realized automatically using machine learning, deep learning techniques or other conventional methods [5]. Machine learning is based on the human classification of different types of images, while deep learning extracts features directly from images. In deep learning, the Convolution Neural Networks (CNNs) are used to make predictions. Such networks have recently achieved high accuracy in image recognition applications, in some cases even outperforming humans [1]. On the other hand, thousands of images are needed to gain sufficient accuracy using deep learning techniques. As a consequence, this causes the learning process to be time-consuming, even if Graphics Processing Units (GPUs) are used.

In this paper, the Deep Learning Neutral Networks (DLNNs) designed for a Biomimetic Underwater Vehicle (BUV) [6] are presented. Biomimetic means that the vehicle can reproduce fish-like behaviour [7] or imitate other marine animals like a seal [8]. This kind of underwater vehicle can confidentially inspect fauna and flora or perform a hidden approach if a military application is considered. For underwater detection of mines [9], exploration of shipwrecks or observation of the process of corrosion of barrels with chemical residues after World War II, autonomous navigation as well as an autonomous control system is definitely desirable. We focused on the Baltic Sea, an area with hundreds of tons of ammunition abandoned after WWII, and an area of many wrecks, including a fuel tank with an unknown technical condition. Because of an erosion [10], their technical conditions should be periodically monitored in order to avoid ecological disaster. In addition, strong sea currents impede access to many places or make inspection very dangerous and risky to divers' health and lives. When exploring the wreckage of the shipwreck using a remote-controlled underwater vehicle, there is a high risk of having a blocked cable. Therefore, the underwater vehicle should be upgraded for autonomous missions.

Due to the strong attenuation of the electromagnetic wave in water, the passive and active hydroacoustic system and the vision system was used for surrounding environment observation. The hydroacoustic passive system for moving obstacles avoidance was depicted in [11]. An underwater image gathered using a video camera is presented in Figure 1, while the sonograms from the sonar system are depicted in Figure 2. A video system is used for passive observation at close range, while sonar images are used to observe objects over greater distances. The visibility range of a vision system strongly depends on the environmental conditions [12] observed in various water reservoirs: distortion of light, light scattering and filtration, luminosity, presence of flares and water turbidity. The sonar system can only register the shape of the scanned object, while the vision system can provide more information about the technical condition of the observed object. For the image contrast and color cast improvements, an underwater image restoration approach based on a parallel CNN and the underwater optical model was proposed in [13]. The deep learning for underwater image recognition in small sample size situations was discussed in [14]. There are also attempts to use sonar data. The paper [9] tackles the problem of automatic detection and classification of underwater mines on images generated by a Synthetic Aperture Sonar (SAS) using the DLNN. In [15], the recognition model of a shipwreck target using side-scan sonar and CNN is presented. Although the sonar imagery has a lower resolution or grey-scale color, the corrected classification of sonar imagery for jellyfish detection presented in the paper [16] is improved by up to 90%.

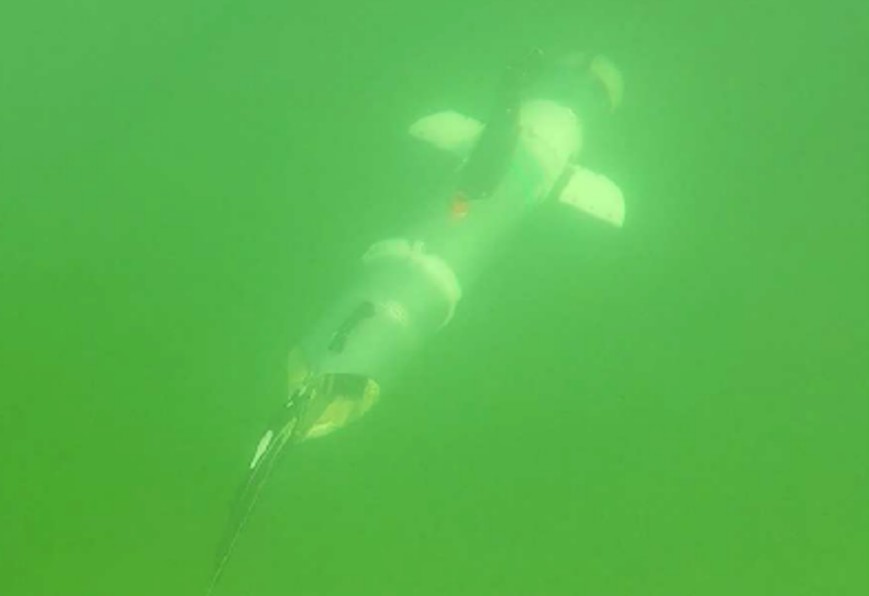

**Figure 1.** Image of the autonomous underwater vehicle achieved from vision system.

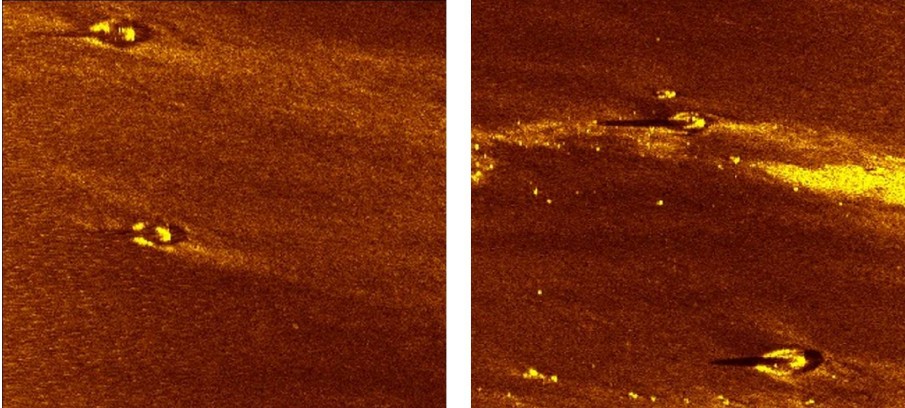

**Figure 2.** Sonograms of underwater mines achieved from side scan sonar.

In Figure 3, the photo of the biomimetic autonomous underwater vehicle before immersion is presented. This photo was taken during the final demonstration of the international project at the European Defence Agency (EDA) under the name SABUVIS at the end of 2018. This vehicle is equipped with different sensors, communication and navigation systems. Further, the video camera is marked in red, the hydroacoustic sensors from a passive system are marked in green, and the side sonar system sensor is marked in white. Applied sensor systems can record environmental information in an image format suitable for deep network analysis. The underwater obstacle avoidance system [11] can be supported after implementing DLNN in the BUV control system. Short prediction time is needed to meet the requirements for vehicle motion control time [17]. The challenges of anti-collision tasks in a water environment were presented in [18], but the test was provided for a small autonomous surface vehicle using radar sensors. When choosing a DLNN type [19], the following factors should be considered: accuracy, size and prediction time. Pretrained means that image classification network has already been trained and is ready to use into a new task, after transfer learning (Figure 4). Transfer learning consists of taking features learned in one problem and leveraging them on a new, similar problem. Usually, transfer learning consists of the following stages: (1) Take layers from a previously trained neural net, (2) Replace some layers especially classification layer, (3) Train the new layers on your dataset. Re-training on the new dataset is usually made with a low learning rate. Subsequently, this can potentially achieve desired improvements, by incrementally adapting the pretrained net to the new data.

The DLNNs comparison presented in [20] was made according to results of the ImageNet Large-Scale Visual Recognition Challenge (ILSVRC) [21] based on the ImageNet database [22]. All presented networks [20] have the same input image (RGB) format. However, the same DLNN trained in different ways can achieve different accuracy depending on the parameters used during the training. In this paper, the next DLNNs were used and compared in object classification for underwater purposes:

- AlexNet [21];
- DenseNet-201 [23];
- GoogLeNet [24];
- Inception-ResNet-v2 [25,26];
- Inceptionv3 [27];
- MobileNetV2 [28];
- NASNet-Mobile [29];
- ResNet-18 [30];
- ResNet-50 [30];
- ResNet-101 [30];
- ShuffleNet [31];

-	SqueezeNet [32];
-	VGG-16 [33];
-	VGG-19 [33];
-	Xception [34].

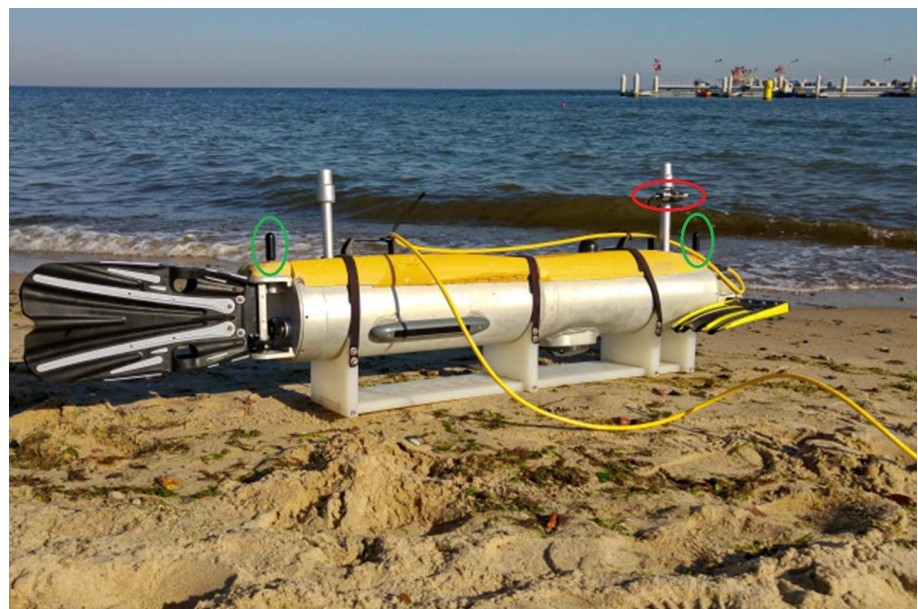

**Figure 3.** The Biomimetic Underwater Vehicle no. 2 (BUV2) before launching (the video camera is marked by a red circle and the hydrophones are marked by a green circles).

The research question undertaken in this paper is how effective can pretrained DLNN for OCUV be in the Baltic Sea environment. In order to answer this question correctly, the best structure and parameters of DLNN should be guaranteed after its training process. Also, to obtain this condition, three different training methods for AlexNet were compared, and the training options of these methods were selected each time automatically by the Genetic Algorithms. Then, fifteen of the most promising DLNNs were used and the average values of training accuracy, verification accuracy and the average training time of a single network were presented. In this paper, only four following classes of underwater objects were assumed: divers, fish, underwater vehicles, water. The last class 'water' is destined for rejection of video frames, not including the objects belonging to the first three classes. During a mission when the network processes many video images, only frames classified to one of the first three classes are to be recorded. Therefore, the layer was modified to the fully connected layer with four outputs (Figure 4). When there are not enough labeled samples, transfer learning should be used. To obtain the final form of DLNN, different training methods were adopted. Moreover, proper values of training options were selected to achieve adequate accuracy. Therefore, three different training methods were compared in the project for the selection of the best training method for the OCUV problem in the Baltic Sea. The training options of all the methods were optimized using GA. The whole research is time-consuming, and it needs a calculation platform with several GPUs.

This paper is organised as follows: in the following section, the state-of-the-art is presented. Then, the research problem is formulated. Next, the data collection process is described. In the next section, the training methods of the pretrained DLNNs as well as the genetic algorithms used to find the best values of training options are presented. Last, this paper addresses the result and gives the conclusion, which includes the direction of future research.

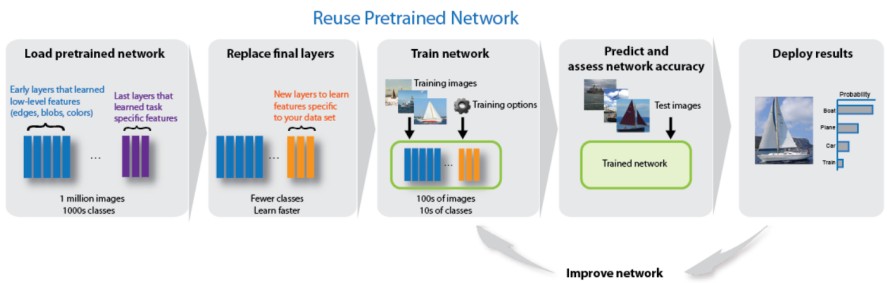

**Figure 4.** The scheme of Pretrained DLNN transfer learning [20].

## 2. State-Of-The-Art

The process of learning DLNN can be supervised [35], semi-supervised [36] or unsupervised [37]. Deep learning architectures can be deep neural networks, deep belief networks and recurrent neural networks [19,38]. After winning the competition in the ILSVRC in 2012, the ALEX network became the most popular one, and new networks are often compared with its achievements. The same accuracy as AlexNet, but with fifty times fewer parameters, was achieved with SqueezeNet. Moreover, the compression techniques allowed the memory size to be reduced to 0.5 MB [31] compared to 240 MB for AlexNet. For low power consumption and strong computing capability, the use of Field-Programmable Gate Array (FPGA) for an image recognition system on the Convolution Neural Network [39] can also be considered. A sufficiently small model could be stored directly on the FPGAs, which have often less than 10MB of on-chip memory. In Table 1 a number of layers, size, and a number of parameters for all analyzed DLNNs are depicted. Early designed DLNNs such as AlexNet, VGGNet, GoogleNet and ResNet were improved in training algorithms during the next years. The Dense Convolutional Network (DenseNet) was introduced in [23] with the connection between each layer to every other in a feed-forward fashion. It was made due to the restriction of shorter connection between layers close to the input and those close to the output. ShuffleNet and MobileNetV2 were designed for devices with small amounts of high-speed software controlled cache memory. Only NASNet-Mobile network does not consist of a linear sequence of modules, while DenseNet-201 has the most significant number of layers with 77MB of memory space. The number of layers generally improves efficiency, but a learning process is then more complicated. Some networks such as ResNets and DenseNets are quite similar, but their behaviour varies greatly. The difference is mainly because of nonlinear transform inputs in each layer, i.e., instead of summation (ResNets) they are concentrated (DenseNets).

**Table 1.** DLNNs Parameters.

| Name of DLNN | Depth | Size | Parameters (Milions) | Image Input Size |
|---|---|---|---|---|
| AlexNet | 8 | 227 MB | 61 | 224-by-224 |
| vgg16 | 16 | 515 MB | 138 | 224-by-224 |
| vgg19 | 19 | 535 MB | 144 | 224-by-224 |
| SqueezeNet | 18 | 4.6 MB | 1.24 | 227-by-227 |
| GoogLeNet | 22 | 27 MB | 7 | 224-by-224 |
| Inceptionv3 | 48 | 89 MB | 23.9 | 299-by-299 |
| DenseNet-201 | 201 | 77 MB | 20 | 224-by-224 |
| MobileNetV2 | 53 | 13 MB | 3.5 | 224-by-224 |
| ResNet-18 | 18 | 44 MB | 11.7 | 224-by-224 |
| ResNet-50 | 50 | 96 MB | 25.6 | 224-by-224 |
| ResNet-101 | 101 | 167 MB | 44.6 | 224-by-224 |
| Xception | 71 | 85 MB | 22.9 | 299-by-299 |
| Inception-ResNet-v2 | 164 | 209 MB | 55.9 | 299-by-299 |
| ShuffleNet | 50 | 6.3 MB | 1.4 | 224-by-224 |
| NASNet-Mobile | - | 20 MB | 5.3 | 224-by-224 |

A comprehensive description of the applications of deep learning for underwater image analysis in recent years is described in [40]. It was concluded that there is developing automation in the analysis of digital seabed imagery using the DLNN for detecting and classifying various underwater marine objects, especially sea-grass, meadows and coral reefs. In the paper [41], an objective classification approach for high-resolution remote sensing imagery using among others DLNN was described. A system for real-time jellyfish monitoring from underwater video recordings presented in [40] uses a deep object detection neural network to detect and classify jellyfish instances, combined with a quantification algorithm. The presented system was planned to be implemented at a floating station and executed online; however, the final system has not been implemented into the floating platform yet.

## 3. Research Problem

Regarding operation in the Baltic Sea, there are problems with open access to a database with a larger number of images registered in this environment. Therefore, following tests in the real environment, periodic DLNN training was assumed. Thus, this explains why resorting to obtaining our images. Some initial research has already been carried out to evaluate the effectiveness of using pretrained AlexNet DLNN in OCUV. Firstly [17], the Neural Network Toolbox with DLNN included in Matlab 2019 [20] was examined. During the initial tests [42], three different training algorithms with different discrete values of training options of the pretrained AlexNet DLNN were examined. For training and verification processes of the DLNNs, 50 different images from each of the object category were used in the following way: 70% of all the images were randomly selected for training and the rest of the images were used for verification process. The DLNNs training and verification processes consist of 50 different images from each object category; further, 70% of all images were selected randomly for training while the rest of the images were used for the verification process.

To obtain statistical results, the training and verification processes with random selection of the training and verification images at the beginning of the training were repeated 30 times. Quite promising results were obtained, i.e., the mean verification accuracy in 30 trails above 90% were achieved for selected variants of the DLNN [17]. After that, the total number of images was increased to 450. Additionally, harder to recognize photos were selected. Unfortunately, the results of this research were not as good as previous results. Therefore, the Genetic Algorithm (GA) was applied for selection of training options of the Stochastic Gradient Descent with Momentum optimizer (SGDM) for training DLNN [42]. Finally, the satisfying training and verification accuracy were obtained, i.e., on average, 100% of training images and 94% of verification images were recognized correctly [42]. The drawbacks of the initial tests mentioned above are (1) too few training and verification images that could result in overfitting and (2) no comparison of different methods whose training options have been optimally selected.

The research problem in this paper consists of evaluating the effectiveness of pretrained DLNNs in OCUV for 2400 images (including 450 source images mentioned above and an additional 150 images representing water without any obstacles and 1800 images achieved obtained using image data augmentation). Three training methods (SGDM, RMSProp, Adam) were used in the first stage of research, then Adam method was selected for the remaining tests due to the highest average accuracy and shortest training time. Training options were selected by GA each time. The criteria of the effectiveness evaluation are (1) the accuracy of OCUV understood as a training and a verification accuracies higher than 90%, (2) the rate of training deep neural network understood as the calculation time of server with several GPUs needed to find the DLNN with desired accuracy of OCUV not longer than one week. Each accuracy (training or verification) indicates what part of the images has been recognized, i.e., '1' means that all the images were recognized correctly and '0' means that none of the images were recognized properly. The first criterium should be maximized and the second should be minimized.

To solve this problem, the pretrained DLNNs were applied to start the training process. Moreover, the GA was accepted as an optimization algorithm for searching training options of the compared methods.

The task of the DLNNs was to classify underwater images to one of four object categories: (1) divers, (2) Unmanned Underwater Vehicles UUVs, (3) fish, (4) water. Such classification is connected with the military purpose of the vehicle, i.e., it should recognize underwater creatures, which are harmless, divers, who could be an enemy and or other UUVs, because possible future swarms of UUVs. Due to the small number of photos taken in the Baltic Sea, the final tests were performed in a real environment.

## 4. Data Collection

DLNN processing requires resizing each image to the dimensions of the first layer. Image input size for every analysed neural networks are depicted in the last column in Table 1. The classification accuracy of the underwater image with special image characteristic is lower than the corresponding result of images in the air [13]. It is caused mainly by heterogeneous refraction of light at the water-air interface, i.e., rays of different wavelengths undergo different bending and strong decrease of illumination with depth. In addition, the light is scattered on the water molecules and the rays of light with different wavelengths are absorbed by the water molecules with varying intensity, depending on the depth. Because of these factors, the visibility can change in different oceans and seas. In most cases it achieves 20 m and more, while in the Baltic Sea is in a range of 2 to 8 m. The other difference is colour absorption. In clean water, a total absorption of red colour reaches 5 m, yellow 50 m and green 110 m. The blue colour is suppressed the least. In clean water, a total absorption of blue colour reaches even 275 m. In water reservoirs with good visibility, the underwater images have blue colour, while in the Baltic Sea, where the visibility is low, the underwater images have green colour. One of the methods used to make the images more prominent is the spectrum corrector equalization [43]. The number of images taken in a highly visible aquatic environment is a real problem when preparing DLNN data for the Baltic Sea. Therefore, training data is enriched with new photos during each test in a real environment. Each of the trained network can be implemented in a Jetson TX2 system installed on board of a BUV2 using a Matlab coder. Jetson TX2 equipped with the GPU almost allowed us to receive the result of classification on-line. To avoid off-line operations, only the 5-frames-per-second mode was selected.

To train DLNN and then to verify them, 600 photos were downloaded from the internet: 150 images with divers, 150 with fishes, 150 with UUVs and 150 including only sea water. Using random reflection, rotation and translation, an additional 1800 images have been produced. All the photos include one or two objects for classification, e.g., one or two divers or a single or a swarm of fishes. Figure 5 represents examples of collected images of divers, fishes and an unmanned vehicle. They were taken in different waters, by different photographers and in various scale and numbers. It seems that it will be hard to obtain an optimal solution for a deep neural network, taking into account the fact that some people can also have problems with classification of the images. In total, 70% of the photos were selected randomly for the training, while the remaining 30% were used for the verification. The process of random selection of images were repeated 6 times for achieving average values of inaccuracies obtained in 6 trials during training and verification.

The research was performed on 2400 images (including 450 source images mentioned above and 150 additional images of water without obstacles, and an additional 1800 images were achieved using image data augmentation).

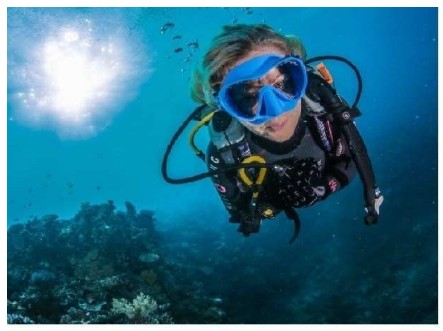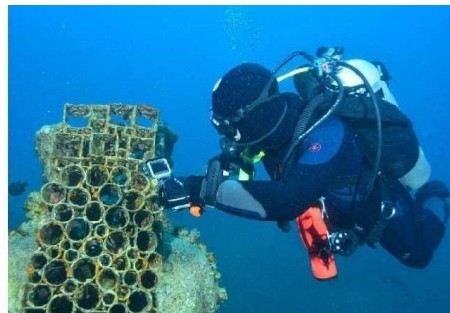

Images courtesy of the Gdynia Dive Centre DTS Piotr Niewiński (www.gdynia.dive.pl)

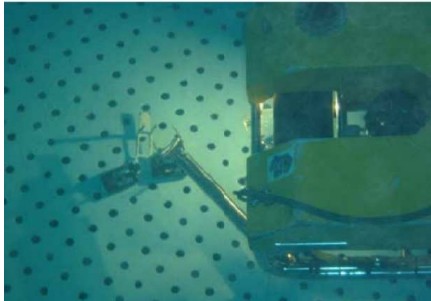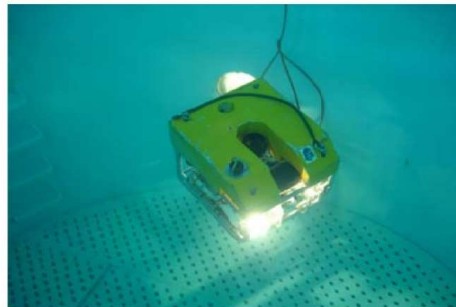

Images courtesy of Prof. Adam Olejnik (Department of Underwater Technology)

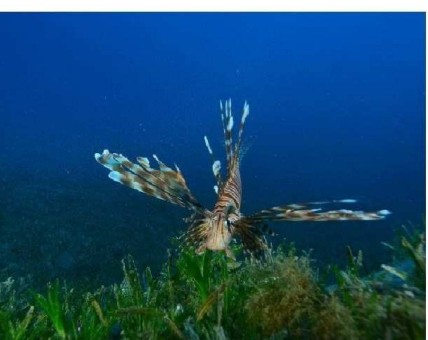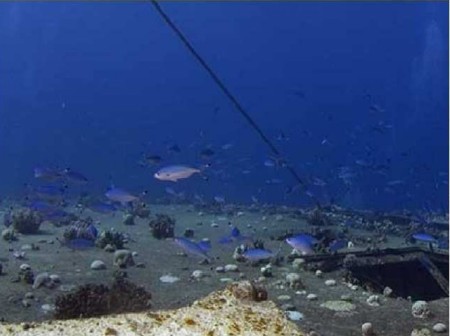

Images courtesy of the Gdynia Dive Centre DTS Piotr Niewiński (www.gdynia.dive.pl)

**Figure 5.** Images from the dataset: divers (first row), UUVs (second row) and fishes (third row).

## 5. Training Methods

This section discusses the structure of the pretrained DLNN modified for the goal of the research on underwater object recognition. Then, the training methods of the DLNN are described. At the end of this section, the description of the GA used for optimization of the training options is presented.

There are different training methods of the DLNN, as described in [44,45]. Here, the following three gradient methods have been used [7,20]:

SGDM—The stochastic gradient descent with momentum optimizer
RMSProp—The root mean square propagation optimizer
Adam—The derived from adaptive moment estimation optimizer.

The optimization process of the training options were established with the genetic algorithm (GA). The following criteria were taken into consideration:

- the accuracy of the underwater objects recognition understood as the sum of training and verification accuracies
- the rate of training deep neural network understood as the time that the calculation server with 6 GPUs needs to find the tuning options using the GA.

In the next paragraph, only the most important mathematical descriptions of the gradient methods parameters are presented.

The SGDM updates the network parameters (weights and biases) to minimize the loss function by taking small steps in the direction of the negative gradient of the loss. The additional momentum factor helps to reduce the oscillation, which may appear along the path of steepest descent towards the optimum [11]. The stochastic gradient descent with momentum algorithm uses a single learning rate for all the parameters. This algorithm is defined as:

$$\theta_{n+1} = \theta_n - \alpha \nabla E(\theta_n) + \gamma(\theta_n - \theta_{n-1}) \tag{1}$$

where:

$n$ is the following steps of iterative process of training,
$\alpha$ is the learning rate,
$\theta$—The vector of trained parameters,
$E(\theta)$ is the loss function,
$\gamma$ is the momentum factor determining how much the previous step influences on the current step of iteration.

The RMSProp uses different learning rates for different parameters and that can automatically adapt to the loss function being optimized. This algorithm is defined as:

$$\theta_{n+1} = \theta_n - \frac{(\alpha \nabla E(\theta_n))}{\sqrt{v_n} + \epsilon} \tag{2}$$

where:

$$v_n = \beta_2 v_{n-1} + (1 - \beta_2)[\nabla E(\theta_n)]^2 \tag{3}$$

where:

$\beta_2$ is the decay rate of the moving average for squared gradient,
$\epsilon$ is the constant higher or equal to zero.

The derived from adaptive moment estimation (Adam) uses a parameter update that is similar to RMSProp with an additional momentum term. The update is calculated based on the following equation:

$$\theta_{n+1} = \theta_n - \frac{\alpha m_n}{\sqrt{v_n} + \epsilon} \tag{4}$$

where:

$$m_n = \beta_1 m_{n-1} + (1 - \beta_1)\nabla E(\theta_n) \tag{5}$$

and

$$v_n = \beta_2 v_{n-1} + (1 - \beta_2)[\nabla E(\theta_n)]^2 \tag{6}$$

The learning rate can be constant or variable during the training process. At the beginning of the training, $\alpha$ is equal to the initial learning rate $\alpha_i$. Then, the value of $\alpha$ can be updated by multiplying with a certain factor called the learning rate drop factor $\alpha_{rdf}$ for every fixed number of epochs, called the learning rate drop period $\alpha_{rdp}$. The length of the training process is limited by the maximum number of epochs $e_{max}$. A single epoch is the full pass of the training algorithm over the entire training set. An iteration is one step taken in the gradient descent algorithm towards minimizing the loss function using a mini-batch. The size of the mini-batch to use for each training iteration, called mini batch size $b_{min}$, is a subset of the training set that is used to evaluate the gradient of the loss function and update the weights [20].

*Ga Settings*

In general, the Genetic Algorithm (GA) is a heuristic search that mimics the process of natural selection. The GA is based on an iterative evolutionary procedure involving selection of genotypes for reproduction based on their fitness, and then introducing genetically changed offspring into the next population. The changes are introduced into the offsprings by means of a mutation, a crossover and other genetic operators. The procedure is finished after achieving satisfactory genotypes (a set of features of an individual) which correspond to the phenotypes with high fitness function (the individual from a population) [46]. The GA used in the previous research is described in details in [42]. In the next part of the subsection only the most important settings of the GA are described.

In order to undertake the search for the optimal values of the training options of all the compared training methods, the initial population was generated using Matlab random generator. The population consisting of 15 individuals was accepted. The individuals in the current generation are estimated using the following fitness function [47]:

$$f_{fit} = (1 - A_{tav}) + (1 - A_{vav}) \tag{7}$$

where: $A_{tav}$, $A_{vav}$—average values of accuracy obtained in $n$ trials respectively during training and verification processes. Each accuracy indicates what part of the images has been recognized, i.e., '1' means that all the images were recognized correctly and '0' means that none of the images were recognized properly.

After calculation of the fitness function, reproduction algorithm creates a member of the next generation. In the reproduction, the following operators were used: rank fitness scaling, stochastic uniform selection function, crossover fraction equal to 0.8, Gaussian mutation function.

Fitness scaling converts the raw fitness scores that are returned by the fitness function into values in a range that is suitable for the selection function. The rank fitness scaling scales the raw scores based on the rank of each individual instead of its score. The rank of an individual is its position in the sorted scores. An individual with rank $r$ has a scaled score proportional to $\frac{1}{\sqrt{r}}$.

The selection function specifies how the genetic algorithm chooses parents for the next generation. The stochastic uniform selection function lays out a line in which each parent corresponds to a section of the line of length proportional to its scaled value. The algorithm moves along the line in steps of equal size. At each step, the algorithm allocates a parent from the section it lands on. The first step is a uniform random number less than the step size. The crossover fraction specifies the fraction of the next generation, other than elite children, that are produced by crossover. The Gaussian mutation function adds a random number taken from a Gaussian distribution with mean 0 to each entry of the parent vector.

During the research, the GA was stopped when the maximum number of 50 generations was reached and/or when no change was detected in the best value of the fitness function for 10 maximum stall generations. A more detailed description of the GA used in DLNN is presented in [47].

## 6. Results

The research was conducted in four stages. In the first stage, only $e_{max}$ and $b_{min}$ were tuned with the constant values of the other parameters selected in previous research [17]. In the second stage, the other training options were tuned by means of the GA selected in the first stage values $e_{max} = 48$ and $b_{min} = 43$. In the third stage, the best DLNN was implemented in Jetson TX2, and verification in a real environment was executed. In the fourth stage of the research, 14 others DLNNs were trained and verified using Adam training method and values of training options used for the AlexNet DLNN. Finally, it allows us to compare 15 different pretrained DLNNs in the OCUV problem for the specific Baltic Sea environment.

To estimate effectiveness of training and validation of DLNN, the average values of accuracy obtained respectively during training $A_{tav}$ and validation $A_{vav}$ in $n$ trials were accepted. Additionally,

the training (learning) time was noted for each of the three training methods. Due to the fairly large calculation time (Table 2), the number of trials $n$ was decreased from 30 used in previous tests [17,42] to 6. In the first phase of the second stage, the training options of SGDM were tuned using GA, especially the learning rate drop factor $\alpha_{rdf}$, the learning rate drop period $\alpha_{rdp}$ and the momentum $\gamma$. The tuning process is visualized in Figure 6.

Finally, the following values of the searched parameters were received: the learning rate drop factor $\alpha_{rdf} = 1$, the learning rate drop period $\alpha_{rdp} = 1$, the momentum $\gamma = 0.0625$ with the average training accuracy equal to 1 and the average verification accuracy was equal to 0.932.

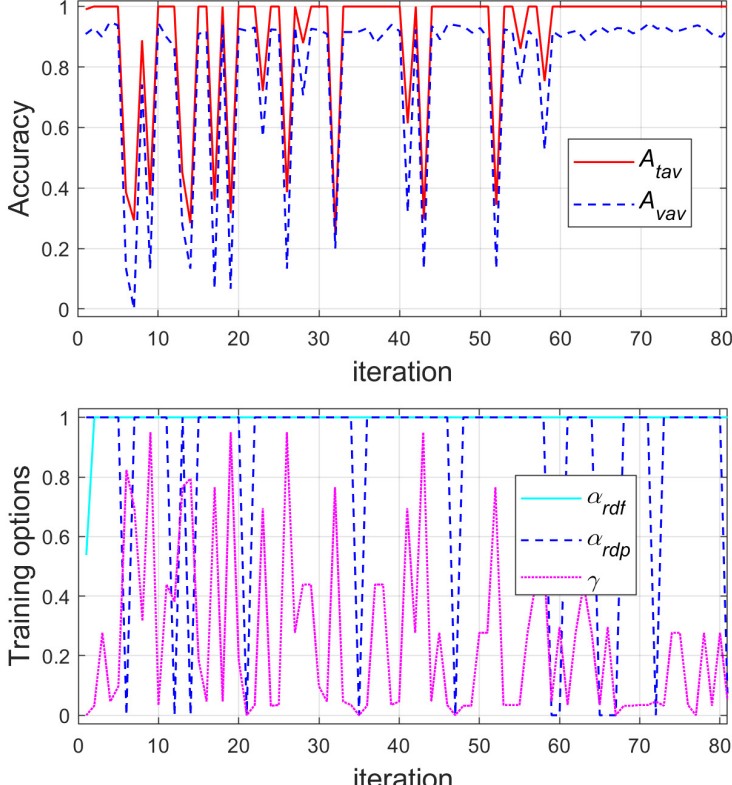

**Figure 6.** Changes of the average values of training $A_{tav}$ and verification $A_{vav}$ accuracies and the learning rate drop factor $\alpha_{rdf}$, the learning rate drop period $\alpha_{rdp}$ and the momentum $\gamma$ of the SGDM during the first 80 iterations of GA.

In the second phase of the second stage of the research, the training options of RMSProp were optimized by GA, especially $\alpha_i$ and $\beta_2$. The default value of $\epsilon = 1 \times 10^{-8}$ was accepted. The constant $\epsilon$ is a denominator offset, which is needed to avoid division by zero in Equation (4). The training process of the RMSProp is illustrated in Figure 7. At the end of this process, the following values of the searched parameters were obtained: the initial learning rate $\alpha_i = 1.33 \times 10^{-4}$, the squared gradient decay rate $\beta_2 = 0.874$ with an average training accuracy equal to 1 and average verification accuracy equal to 0.859.

In the third phase of the second stage of the research, the training options of Adam method were tuned. The training process is visualized in Figure 8.

During this phase, the following three training options were searched by the GA: the initial learning rate $\alpha_i$, the squared gradient decay rate $\beta_2$ and the gradient decay rate $\beta_1$. At the end of this process, the following values of the searched parameters were obtained: the initial learning rate $\alpha_i = 2.23 \times 10^{-4}$, the squared gradient decay rate $\beta_2 = 0.992$, the gradient decay rate $\beta_1 = 0.852$ with an average training accuracy equal to 1 and average verification accuracy equal to 0.911.

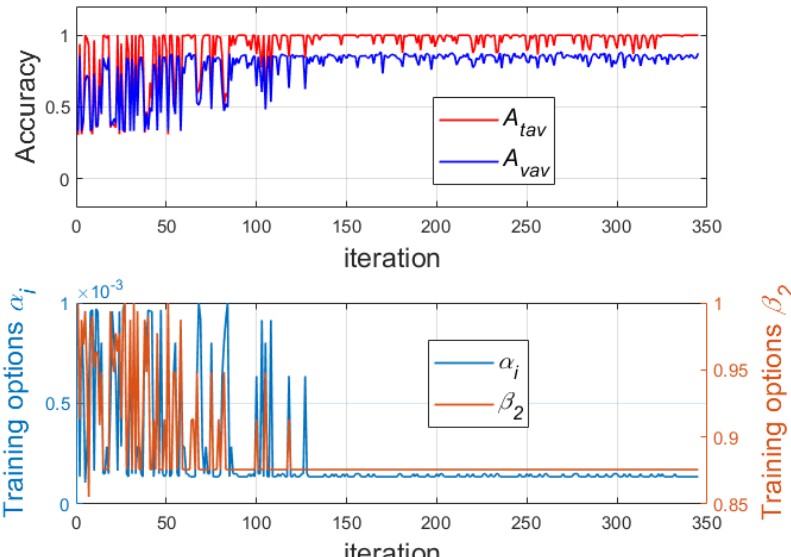

**Figure 7.** Changes of the average values of training $A_{tav}$ and verification $A_{vav}$ accuracies and the initial learning rate $\alpha_i$ and the squared gradient decay rate $\beta_2$ of the RMSProp during the first 150 iterations of GA.

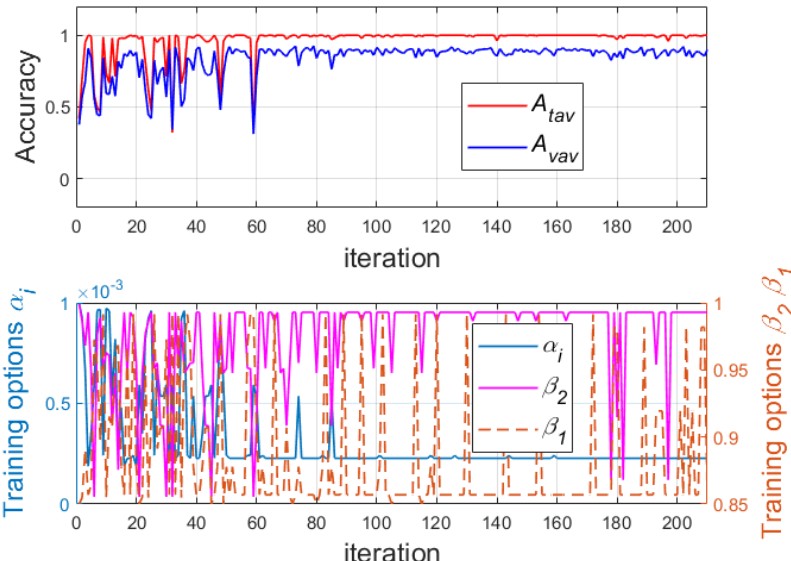

**Figure 8.** Changes of average values of training $A_{tav}$ and verification $A_{vav}$ accuracies and the initial learning rate $\alpha_i$, squared gradient decay rate $\beta_2$ and gradient decay rate $\beta_1$ of the Adam during the first 80 iterations of GA.

Table 2 summarizes all the research results both average values of accuracies obtained respectively during training $A_{tav}$ and validation $A_{vav}$ in *n* trials and the training for each of the three training methods. The results are discussed in detail in the next section.

In Figures 9 and 10 processed images obtained on the following convolutional layer outputs in addition to the source image are shown. Each output of the convolutional layer represents a different feature of the source image. In Figures 9 and 10 only the outputs with the strongest activation were visualized to show which features the network learns to classify in underwater videos.

**Table 2.** The results of SGDM, RMSProp and Adam methods of training DLNN in the OCUV problem.

| Training Methods | Average Training Accuracy [-] | Average Verification Accuracy [-] | Training Time [h] |
|---|---|---|---|
| SGDM | 1 | 0.932 | 218.2 |
| RMSProp | 1 | 0.859 | 429.1 |
| Adam | 1 | 0.911 | 98.5 |

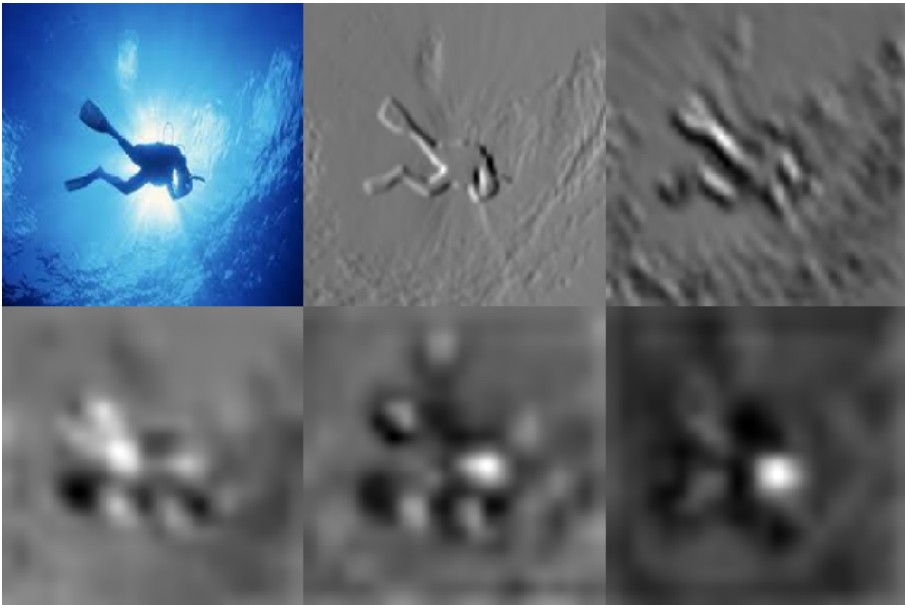

**Figure 9.** Diver wrongly classified as a fish (from top right to bottom left): source image, proccessed images in grey on the following layer outputs of DLNN: conv1, conv2, conv3, conv4 and conv5.

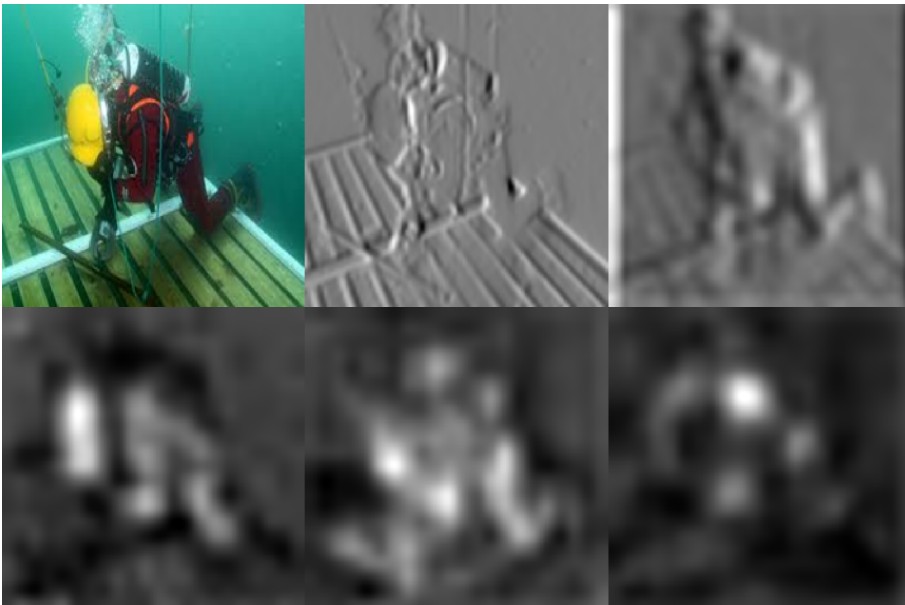

**Figure 10.** Diver wrongly classified as an UUV (from top right to bottom left): source image, processed images in grey for the following layer outputs of DLNN: conv1, conv2, conv3, conv4 and conv5.

The AlexNet trained by SGDM obtained the best accuracy. Therefore, this DLNN was implemented in Jetson TX2 and verified in a real environment. During a real test, 24 objects were classified as divers, fishes or UUVs. The worst result of classification was received for divers. This was probably connected with the fact that there were no images with divers in green background

characteristic for the Baltic Sea in the training data. All the UUVs were recognized correctly even the BUV which was similar to the fish (Figure 11). In addition, after the real test, two other DLNNs trained by RMSProp and Adam methods were verified offline using images recorded by BUV2. The results of test in the form of average accuracies are included in Table 3. As we can see, the accuracy decreases by almost 30%. After training the AlexNet DLNN again using the Adam method and additional images recorded during the real test, the accuracy was increased to the average for 6 trails value 0.917, i.e., even a little more than previously. This indicates that this DLNN can be easily retrained after receiving additional data.

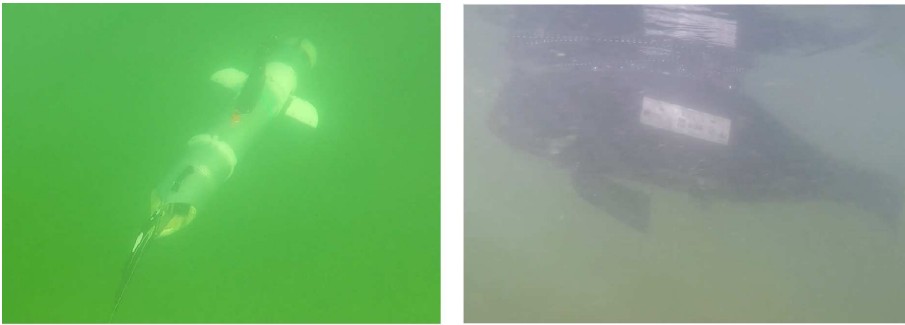

**Figure 11.** BUVs participating in real test of OCUV in the Baltic Sea.

**Table 3.** The results of DLNNs trained using SGDM, RMSProp and Adam methods in the real test in the Baltic Sea.

| Training Methods | Average Accuracy [-] |
|---|---|
| SGDM | 0.650 |
| RMSProp | 0.633 |
| Adam | 0.741 |

In the final stage of research, the comparison of 15 pretrained DLNNs including mentioned above the AlexNet has been carried out. The type of the tested DLNNs and the results of their operation in the form of average in 30 trials training accuracy, verification accuracy and time needed for training single net are included in Table 4. For statistical evaluation of the pretrained networks, each DLNN was trained and verified 30 times using random sets of underwater images selected in the same way as in previous stages of the research. For all the DLNNs, the same training method (Adam) with training options achieved in the previous stage of the research was used. During training process of part of the pretrained networks, GPU low memory warning has been noticed (see 'Remarks' column in Table 4). It is worth underlying that the warning can indicate lower performance of GPU due to additional data transfers with main memory during training of some of the DLNNs. It should be taken into consideration during comparison of calculation time needed for training the DLNNs.

It is worth mentioning that all the tested DLNNs have different number of outputs than 4. In most cases they are able to classify to one of 1000 classes. Therefore, the DLNNs should be adapted for an OCUV as it was shown for the AlexNet, especially the last learnable layer and the final classification layer use to classify the input image should be replaced. In the research, the MATLAB function "findLayersToReplace" was used to find the correct names of these layers and to replace them.

**Table 4.** Comparison of different DLNNs.

| Type of DLNN | Average Training Accuracy [-] | Average Verification Accuracy [-] | Average Training Time of Single Net [h] | Remarks |
|---|---|---|---|---|
| AlexNet | 0.998 | 0.871 | 0.18 | – |
| DenseNet-201 | 1 | 0.979 | 10.16 | GPU low memory |
| GoogLeNet | 1 | 0.937 | 0.18 | – |
| Inception-ResNetV2 | 0.995 | 0.949 | 1.34 | GPU low memory |
| InceptionV3 | 1 | 0.952 | 0.50 | GPU low memory |
| MobileNetV2 | 1 | 0.966 | 0.36 | – |
| NASNetMobile | 1 | 0.958 | 1.87 | – |
| ResNet-18 | 1 | 0.971 | 0.21 | – |
| ResNet-50 | 1 | 0.964 | 0.37 | – |
| ResNet-101 | 0.997 | 0.899 | 0.94 | – |
| ShuffleNet | 1 | 0.953 | 0.29 | – |
| SqueezeNet | 0.999 | 0.931 | 0.19 | – |
| VGG-16 | 0.938 | 0.717 | 1.36 | GPU low memory |
| VGG-19 | 1 | 0.695 | 3.81 | GPU low memory |
| Xception | 1 | 0.961 | 0.51 | GPU low memory |

## 7. Discussion

Based on previous research [17], including a comparison of computing time needed by single CPU i7 and single middle class GPU for training AlexNet DLNN, it can be said that it would not be possible to obtain the presented results of research without using a multi GPUs calculation platform. Former research indicated an almost 20-times faster execution of a numerical research by single GPU than CPU. A satisfactory result was also obtained for mobile microprocessor system Jetson TX2. This enables video processing at 5 frames per second. It is worth mentioning that even when a single image may be misclassified, with the processing rate 5 fps some frames may be classified correctly.

Comparing obtained values of the accuracy, especially the verification accuracy, the SGDM proved to be the best training method in the problem of OCUV. However, the results of accuracy received by the Adam method were also very good, i.e., they were only approx. 2% lower than for the SGDM and they were better for real testing. The RSMProp training method does not exceed the assumed threshold of 90%.

Taking into consideration the learning rate using different training methods, it can be stated that the Adam method is the fastest. Comparing training times (Table 2), it can be seen that the server equipped with six medium class GPUs requires almost four days of calculation using the GA to find the optimal values of Adam methods' training options and almost eight days using the GA to find optimal training options for the SGDM.

Some of the images were not classified correctly. In Figure 9 there is an example of diver's image which was wrongly classified as a fish, and in Figure 10 there is diver's image classified as an UUV.

The obtained verification accuracy of the AlexNet higher than 90% proved effectiveness of using DLNN pretrained on non underwater images for the OCUV problem. This can be further confirmed by looking at examples of images which were classified wrongly (Figures 9 and 10). Some people might also make mistakes by looking at these images. Looking at the processed images on the first convolutional layer, it can be seen that the feature chosen by the network is similar to the one that a human would choose.

As we can see in Table 4, the AlexNet is not the best DLNN for an OCUV, i.e., the other pretrained networks obtained better average training and verification accuracies. The highest verification accuracy was received for the DenseNet-201. This DLNN is quite complicated (201 layers), therefore its training process needs the longest time. Althought, there are several DLNNs with similar accuracy and much shorter time of their training, e.g., ResNet-18, MobileNetV2, ResNet-50, ShuffleNet, GoogLeNet, SqueezeNet. Moreover, these networks does not cause problems connected with GPU low memory warning.

## 8. Conclusions

The aim of the work was to present the results of the project focused on the use of pretrained DLNN for OCUV in an AUV. During this project, methodology was designed to receive OCUV for a specific Baltic Sea environment. In the opinion of the authors, the results should help other researchers trying to use pretrained DLNN for OCUV.

The research on BUVs was continued and the results obtained in this paper were implemented in BUV no. 2 (Figure 3) in order to increase its autonomy. The obtained DLNN was implemented in the Nvida Jetson TX2 hardware using the Matlab GPU coder. The hardware is mounted inside the BUV2. This allows us to make research on an OCUV in a real environment using a series of images of classified objects. Such research demands more time and sea trails, but it gives us the opportunity to continue the learning process of the DLNN and to achieve more efficient DLNN.

Carried out comparison of 15 pretrained DLNNs [20] allowed us to select pretrained networks which are more efficient for an OCUV in specific Baltic conditions taking into consideration both classification accuracy and time consumption.

Future research on OCUV problem will be conducted to search structure of own DLNN using optimization methods, e.g., Pareto search, Particle Swarm Optimization, coevolutionary methods, etc. Regarding usage of other source of underwater image mounted on board of an AUV or the BUV, i.e., a sonar, the research on detection and classification of an unexploded ordnance (UXO) such as mine-like objects using DLNNs is in progress.

**Author Contributions:** Conceptualization, P.S. and P.P.; methodology, P.S. and P.P.; software, P.S. and K.N.; validation, P.S. and K.N.; numerical research, P.S. and K.N.; writing—original draft preparation, P.S. and P.P.; writing—review and editing, P.S. and P.P.; visualization, P.S, P.P. and K.N. All authors have read and agreed to the published version of the manuscript.

**Funding:** This research was funded within statutory grant entitled 'A vision system for analyzing the movement and position of objects in the marine environment' carried out in Polish Naval Academy.

**Acknowledgments:** The results of research presented in the paper were initiated in the European Defence Agency project category B called SABUVIS. The project was carried out by the consortium consisting of the following scientific and industrial Polish partners: Polish Naval Academy AMW—the leader, Cracow University of Technology PK, Industrial Institute of Automatics and Measurement PIAP, Forkos Company, German partners: Bundeswehr Technical Center for Ships and Naval Weapons WTD 71 in Eckernförde, Fraunhofer Institute, and Portuguese partners: R and D center CINAV, Engineering Faculty from Porto University and OceanScan company.

**Conflicts of Interest:** The authors declare no conflict of interest.

## Abbreviations

The following abbreviations are used in this manuscript:

| | |
|---|---|
| Adam | derived from Adaptive moment estimation optimizer |
| AUV | Autonomous Underwater Vehicle |
| BUV | Biomimetic Underwater Vehicle |
| DLNN | Deep Learning Neural Network |
| GA | Genetic Algorithm |
| OCUV | Object Classification in Underwater Video |
| RMSProp | Root Mean Square Propagation optimizer |
| SGDM | Stochastic Gradient Descent with Momentum optimizer |

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
