# Peer review of "The Effectiveness of Using a Pretrained Deep Learning Neural Networks for Object Classification in Underwater Video"

_remotesensing, doi:10.3390/rs12183020_

Round 1

Reviewer 1 Report

This is an excellent paper and all my previous comments have been fully addressed.

Personally I felt the original manuscript contained enough publishable material even without the 15 additional networks. In future it could be interesting to expand on these new results in another paper, examining their strengths and weaknesses in more detail including whether different networks performed better with different hyerparameters.

I wish you the best of luck with your further research and publishing. 

Author Response

We are very grateful for your good estimation of our work. Thank you for your work on our paper.

Reviewer 2 Report

- The article's main concept(s)

Marine robotic vehicle uses vision and sonar underwater sensor to perceive the seafloor environment. Underwater vision sensor can give high quality images but lack range and are easily blur by sediments and depend on water visibility range. Acoustic sonars are good on range and can give us hundreds of meters ranges but their targets resolution are poor. Depending on the mission of the unmanned underwater vehicle it can use vision and sonar systems and gain the best of both systems.

Unmanned underwater vehicle are used on academic, industrial, and military scenarios. From underwater seafloor mapping, oceanographic/hydrographic missions, support to offshore operations, and military operations, such as, mine hunting.

This paper presents an Autonomous Underwater Vehicle that uses bio-inspired robotics solution to mimic fish movements/attitudes. The BUV (Biomimetic Underwater Vehicle) is used by the Polish Naval Academy to detect underwater mines (mine warfare and countermeasures). The current vehicle was developed and used during the EDA project SABUVIS. This stealth or camouflage AUV mimics fish and it is used on Baltic sea to target old World War II ammunition on the seafloor.

The paper focus is on the of use of Deep Learning techniques to underwater video to detect and classify objects on the seafloor. The AUV uses underwater cameras and acoustic sensors. The underwater images are used to create a dataset that is used to train a Deep Learning Neural Network. The authors tested and compared several DLNNs for object identification.

- Overall Comment

In overall,

 This document shows a reasonable understanding of deep learning and computer vision techniques applied to underwater video to detect and classify seafloor objects. The work has a reasonable theoretical base, using useful references and information of general knowledge. The work backbone is the use of pretrained deep learning neural networks for object classification in underwater video implemented on an AUV.

The manuscript has a good structure and it is easy to read. The English is good. But there are some places were the manuscript needs minor English revision.

It is a great marine robotics development that is important to test in live scenarios. Naval exercises such as Portuguese Navy REP(MUS)2019 are a great opportunity. REP exercise, under Maritime Unmanned Systems guidance, allows NATO navies to test their new developments on maritime scenarios.

- Weak and Strong points

Strengths

  • Underwater video for seafloor object classification – deep learning techniques;
  • Great knowledge about deep learning and machine learning;
  • Deep Learning Neural Network;
  • Bio-inspired AUV;
  • Unmanned systems for Naval mine warfare;
  • Genetic algorithm tuning DLNN parameters;
  • Automatic detection and classification of underwater mines.

Weakness

  • Not a big novelty. Uses common sense methods applied together;
  • A sequel to an old work/study by the authors;
  • English needs some revision (minor);

Author Response

Thank you for review of our paper. We are aware that our article is not a big novelty in an area of Deep Learning Neural Networks (DLNNs), because we used pretrained DLNNs in our research, but we hope that our work will help others scientists who are interested in classification of underwater object using video images in specific conditions of Baltic Sea. This work is a finalization of the previous research in this area. The text of the paper has been additionally checked by English language expert to avoid minor language mistakes.

Reviewer 3 Report

Instead of "images achieved using image data augmentation"  better "images obtained using image data augmentation".

Instead of "images were recognized properly" better "images were recognized correctly".

Not clear what "heterogeneous refraction of light at the water-air interface" means.

Statement "water color is absorbed by the water molecules" is formally incorrect. Same goes to "absorption of ... colour".

Reference 44 has no mention of "spectrum corrector equalization".

Eqn. 2 lacks a closing bracket.

Below Eq. 3 it is stated that epsilon is a constant less or equal to zero. However, below epsilon is chosen as 1.e-8.

Not clear why absence of images "with green background" impedes model training - color is not used in model generation.

It is worth mentioning that even when a single image may be misclassified, with the processing rate 5 fps some frames may be classified correctly.

Author Response

We are very grateful for your work on our paper and sent suggestions. All your suggestions have been introduced into revised manuscript.

We have answered each of your points below.

  • Instead of "images achieved using image data augmentation" better "images obtained using image data augmentation".

According to the remark, it has been corrected.

  • Instead of "images were recognized properly" better "images were recognized correctly".

According to the suggestion, it has been corrected.

  • Not clear what "heterogeneous refraction of light at the water-air interface" means.

To the not clear expression "heterogeneous refraction of light at the water-air interface" the following additional explanation has been added “i.e. rays of different wavelengths undergo different bending”.

  • Statement "water color is absorbed by the water molecules" is formally incorrect. Same goes to "absorption of ... colour".

The statement "water color is absorbed by the water molecules" has been replaced by the statement “rays of light with different wavelengths are absorbed by the water molecules with varying intensity”.

  • Reference 44 has no mention of "spectrum corrector equalization".

Thank you very much for your kind suggestion and sorry for our carelessness. We admire your care and rigor, we have replaced reference 44 for the appropriate one.

Instead of:

Jiang,  B.;  He,  J.;  Yang,  S.;  Fu,  H.;  Li,  T.;  Song,  H.;  He,  D.   Fusion of machine vision technology and AlexNet-CNNs deep learning network for the detection of postharvest apple pesticide residues. Artificial Intelligence in Agriculture, 2019,1, 1–8

we put:

  1. Abdullah-Al-Wadud, M. H. Kabir, M. A. Akber Dewan and O. Chae, "A Dynamic Histogram Equalization for Image Contrast Enhancement," in IEEE Transactions on Consumer Electronics, vol. 53, no. 2, pp. 593-600, May 2007, doi: 10.1109/TCE.2007.381734.

  • 2 lacks a closing bracket.

According to the remark, closing bracket has been inserted.

  • Below Eq. 3 it is stated that epsilon is a constant less or equal to zero. However, below epsilon is chosen as 1.e-8.

The expression “epsilon is a constant less or equal to zero” has been replaced by the expression “epsilon is a constant higer or equal to zero”. We apologize for the misleading. In fact, epsilon is a denominator offset, specified as a positive scalar. The solver adds the offset to the denominator in the network parameter updates to avoid division by zero.

  • Not clear why absence of images "with green background" impedes model training - color is not used in model generation.

The images given on the input of the DLNN are processed by the following layers to obtain the best features of the images to make the final classification. The color of the each pixels is given on the input and then processed into the desired information.

Results of the research indicates that the images with green background which had not been used for training they were not recognized correctly.

  • It is worth mentioning that even when a single image may be misclassified, with the processing rate 5 fps some frames may be classified correctly.

The remark has been inserted in the text to make additional explanation. Thank you for this suggestion.

This manuscript is a resubmission of an earlier submission. The following is a list of the peer review reports and author responses from that submission.

Round 1

Reviewer 1 Report

Although this work is a project report, the technical contribution is negligible. Latest works on underwater object detection and recognition have not been discussed in this work. Also, the deep learning model of AlexNet is first reported in 2012 which is 8 years ago, since then, lot of more competitive deep learning frameworks such as ResNet, DenseNet, Inception-ResNet, DPN for image recognition are proposed and proved more effective than AlexNet. These works are not fully discussed in this work.

This work would be more suitable for a short conference presentation rather than a journal paper.

Reviewer 2 Report

The paper is very interesting and provides a thorough description of deep learning training strategies for a novel underwater application. The onboard results are impressive and the scope for future work when combined with more 'real world' training data is exciting.

Some very minor comments as follows. 

  • Whilst both are correct, I think 'fish' rather than 'fishes' would read better in the context of this topic.
  • Line 79, the sentence about pretraining and transfer learning could be expanded to explain this concept further.
  • Line 103, this sentence appears to be incomplete?
  • Line 115, I think it would be preferable to start a new paragraph when switching to discussing the transfer learning strategy, possibly swapping the sentence order with the one at line 121.
  • Section 3, the text could be improved by clarifying that there was a final test carried out in the real environment when discussing the training, validation and testing datasets. The 'verification' at line 170 could be replaced with 'testing'. 
  • Line 158, possibly include a precise definition of 'accuracy' as there are many metrics commonly used in image classification problems.
  • Line 257, is this the population of possible training options? Perhaps a table showing the different varied hyperparameters and their population ranges would help readers who are less familiar with genetic algorithms understand the concept better. 
  • Figure 6 - should there be a second y-axis for the momentum term? 
  • Line 359, is this statement necessary? Is is not expected that the deeper more abstract layers of a CNN are less interpretable to humans? See Zeiler M.D., Fergus R. (2014) Visualizing and Understanding Convolutional Networks.
  • The references appear to be numbered alphabetically rather than in order of appearance, is this correct for the style guide for this special issue?